# Successful Incorporation of Exosome-Capturing Antibody-siRNA Complexes into Multiple Myeloma Cells and Suppression of Targeted mRNA Transcripts

**DOI:** 10.3390/cancers14030566

**Published:** 2022-01-23

**Authors:** Emi Soma, Asako Yamayoshi, Yuki Toda, Yuji Mishima, Shigekuni Hosogi, Eishi Ashihara

**Affiliations:** 1Department of Clinical and Translational Physiology, Kyoto Pharmaceutical University, 5 Nakauchi, Misasagi, Yamashina, Kyoto 607-8414, Japan; ky15176@ms.kyoto-phu.ac.jp (E.S.); tda@mb.kyoto-phu.ac.jp (Y.T.); hosogi@mb.kyoto-phu.ac.jp (S.H.); 2Chemistry of Functional Molecules, Graduate School of Biomedical Sciences, Nagasaki University, 1-14 Bunkyo-machi, Nagasaki 852-8521, Japan; asakoy@nagasaki-u.ac.jp; 3Department of Medical Oncology, Dana-Farber Cancer Institute, Boston, MA 02215, USA; mishima_y@brightpathbio.com

**Keywords:** exosome, CD63, siRNA, nucleic acid medicine, drug delivery systems, antibody, multiple myeloma, hematologic malignancy

## Abstract

**Simple Summary:**

Although nucleic acid medicines are expected to function as new therapeutic agents, their targeted delivery into cancer cells, particularly hematologic cancer cells, via systemic administration, is limited. Based on our previous finding that tumor cell-derived exosomes are preferentially incorporated into their parental cancer cells, we previously demonstrated that anti-CD63 monoclonal antibody (mAb)-oligonucleotide complexes targeting exosomal microRNAs with linear oligo-D-arginine (Arg) linkers (9mer) were transferred into solid cancer cells and inhibited exosomal miRNA functions. To challenge the delivery of siRNAs into hematologic cancer cells, we developed exosome-capturing anti-CD63 mAb-conjugated small interfering RNAs (siRNA) with branched Arg linkers (9+9mer). Anti-CD63 mAb-conjugated complexes were incorporated into multiple myeloma (MM) cells. Moreover, these exosome-capturing mAb-conjugated siRNAs successfully decreased the mRNA transcript levels of targeted mRNAs in the MM cells. This technology could lead to a breakthrough in drug delivery systems for hematologic cancer therapy.

**Abstract:**

Nucleic acid medicines have been developed as new therapeutic agents against various diseases; however, targeted delivery of these reagents into cancer cells, particularly hematologic cancer cells, via systemic administration is limited by the lack of efficient and cell-specific delivery systems. We previously demonstrated that monoclonal antibody (mAb)-oligonucleotide complexes targeting exosomal microRNAs with linear oligo-D-arginine (Arg) linkers were transferred into solid cancer cells and inhibited exosomal miRNA functions. In this study, we developed exosome-capturing anti-CD63 mAb-conjugated small interfering RNAs (siRNAs) with branched Arg linkers and investigated their effects on multiple myeloma (MM) cells. Anti-CD63 mAb-conjugated siRNAs were successfully incorporated into MM cells. The incorporation of exosomes was inhibited by endocytosis inhibitors. We also conducted a functional analysis of anti-CD63 mAb-conjugated siRNAs. Ab-conjugated *luciferase+* (*luc+*) siRNAs significantly decreased the luminescence intensity in OPM-2-luc+ cells. Moreover, treatment with anti-CD63 mAb-conjugated with *MYC* and *CTNNB1* siRNAs decreased the mRNA transcript levels of *MYC* and *CTNNB1* to 52.5% and 55.3%, respectively, in OPM-2 cells. In conclusion, exosome-capturing Ab-conjugated siRNAs with branched Arg linkers can be effectively delivered into MM cells via uptake of exosomes by parental cells. This technology has the potential to lead to a breakthrough in drug delivery systems for hematologic cancers.

## 1. Introduction

Nucleic acid medicines including small interfering RNAs (siRNAs), antisense oligonucleotides, and RNA aptamers have been developed as new therapeutic agents that directly suppress target molecules and control the pathophysiology of many diseases [1,2,3,4]. Although this approach is effective against age-related macular degeneration and muscular dystrophy when these agents are administered locally [2,5], targeted delivery of siRNAs into cancer cells, particularly hematologic cancer cells, via systemic administration is limited by a lack of efficient and cell-specific delivery systems. Therefore, it is essential to develop effective drug delivery systems (DDSs) for tumor-specific delivery.

Multiple myeloma (MM) is a hematologic malignancy characterized by malignant plasma cell proliferation, production of M-proteins, and associated with organ damage, including bone lesion, renal failure, anemia, and hypercalcemia [6]. MM remains incurable, despite the emergence of novel effective agents, including proteasome inhibitors [7], immunomodulatory agents [8], and antibody (Ab)-based medicine [9], and advances in hematopoietic stem cell transplantation conjugated with high-dose chemotherapeutics [10,11]. Hence, the development of further novel agents using various approaches is required.

Exosomes are extracellular vesicles of diameter 50–100 nm, which contain nucleic acids, lipids, and proteins, and control intercellular communication in both healthy physiology and pathophysiology [12,13]. In recent years, treatment strategies using exosomes as a DDS have been investigated [14,15,16]. We previously demonstrated that exosomes exhibit cellular tropism and that exosomes derived from cancer cells are preferentially incorporated into their parental cells [17]. Moreover, integrins on the surface of cancer cell-derived exosomes can determine the organs to which tumors metastasize [18,19]. Based on these observations, we hypothesized that monoclonal antibody (mAb)-conjugated siRNAs that capture exosomes can efficiently deliver siRNAs into cells, and we have developed an Ab-conjugated nucleic acid medicine complex. We have found that anti-CD63 mAb-anti-miRNA oligonucleotide complexes (ExomiR-Trackers) were delivered into Cal27 human oral squamous carcinoma cells and that these Ab-oligonucleotide complexes successfully inhibited the exosomal miRNA functions [20]. In the present study, we have developed a DDS that can capture exosomes released from MM cells using a mAb against a molecule on the exosome surface to target hematologic cancers (Figure 1). We demonstrate that these exosome-capturing anti-CD63 mAb-conjugated siRNA complexes can be successfully introduced into MM cells, where they suppress targeted mRNAs.

## 2. Materials and Methods

### 2.1. Cell Lines

The human MM OPM-2 and NCI-H929 cell lines (Deutsche Sammlung von Mikroorganismen und Zellkulturen GmbH, Braunschweig, Germany), and OPM-2-*lucifease+* (*luc+*) cells [21] were cultured in RPMI 1640 (Wako Pure Chemical Industries, Osaka, Japan) containing 10% heat-inactivated, exosome-depleted fetal bovine serum (FBS; Sigma-Aldrich, St. Louis, MO, USA) and 1% penicillin-streptomycin (PC/SM; Wako Pure Chemical Industries). The human cervical cancer line, HeLa cell (American Type Culture Collection, Manassas, VA, USA), was cultured in Dulbecco’s Modified Eagle’s medium (DMEM) (Wako Pure Chemical Industries) containing 10% FBS and 1% PC/SM. Depletion of exosomes from FBS was performed as previously described [17]. All cell lines were maintained at 37 °C in a fully humidified atmosphere of 20% O_2_, 5% CO_2_, and 75% N_2_.

To confirm the function of Ab-conjugated complexes, we established HeLa-luc cells. Approximately 1 × 10^4^ HeLa cells were transfected with 25 ng of the pGL4.13[luc2/SV40] vector (luc2 reporter gene 499–2151) (Promega, Madison, WI, USA) using Lipofectamine 3000 (Thermo Fisher Scientific, Waltham, MA, USA), according to the manufacturer’s instructions.

### 2.2. Preparation of Antibody-Conjugated Complexes

Anti-CD63 mAb (Human Anti CD63; COSMO BIO Co., LTD, Tokyo, Japan) was used to develop Ab-conjugated siRNA complexes to capture exosomes derived from MM cells. To synthesize antibody-drug conjugates (ADCs), anti-CD63 mAb was bound to siRNA molecules, as previously described [20]. In brief, anti-CD63 mAb was thiolated using Traut’s Reagent in phosphate buffer (pH 8.0) containing 5 mM EDTA at 25 °C for 1 h, and Ab was then conjugated with Cys(Npys)-(D-Arginine)_9_ (9mer, linear Arg) (AnaSpec, Fremont, CA, USA) or Cys(Npys)-(D-Arginine)_9_+(D-Arginine)_9_ (9+9mer, branched Arg) (Peptide Institute, Inc., Osaka, Japan). After completion of the reaction, gel filtration was performed using Zeba Spin Desalting Columns (7K MWCO) (Thermo Fisher Scientific) to remove unreacted Traut’s Reagent. Thereafter, the anti-CD63 mAb-linear (9mer) or branched (9+9mer) Arg was mixed with siRNAs at the indicated molar ratios and incubated at room temperature for 15 min to obtain anti-CD63 mAb-conjugated siRNA complexes. The scheme for the synthesis of anti-CD63 mAb drug conjugate with a polyarginine linker is shown in Figure 2.

Two siRNAs were synthesized to conduct luciferase assays to assess ant-CD63 mAb-conjugated siRNA function: siRNA against *luc2* (for HeLa-luc cells) and siRNA against *luc+* (for OPM-luc+ cells). FITC-labeled *luc2*-siRNA (F-*luc2* siRNA) was used to investigate the cellular uptake of Ab-conjugated complexes by laser scanning microscopy. SiRNAs against *MYC* and *CTTNB1* were synthesized to evaluate the suppression of targeted mRNAs in MM cells using quantitative real-time PCR (qPCR), as both molecules have essential roles in MM pathogenesis [22,23] and are effective therapeutic targets against MM [24,25,26]. SiRNAs against *MYC* (Silencer^®^ Select siRNA s9129 and s9130) were purchased from Thermo Fisher Scientific, and both siRNA #1 and siRNA #2 against *CTTNB1* were designed and synthesized (Gene Design Inc., Osaka, Japan). SiRNA sequences are presented in Appendix A. Silencer Select negative control siRNA #2 (Thermo Fisher Scientific) and siGENOME RISC-free siRNA control siRNA (Horizon, Tokyo, Japan) served as control siRNAs.

### 2.3. Luciferase Assay

We first evaluated the efficacy of anti-CD63 mAb-conjugated siRNA complexes using HeLa cells seeded at 4.5 × 10^3^/well in a 96-well plate. The next day, we generated ADCs by mixing anti-CD63 mAb combined with linear Arg linker and either Silencer Select negative control siRNA #2 (Thermo Fisher Scientific) or *luc-2* siRNA at a ratio of 2.5:1 (750:300 (nM)). Twenty-four hours after treatment with ADCs, cells were transfected with the pGL4 luciferase reporter vector (Promega, Madison, WI, USA) using Lipofectamine 3000 (Thermo Fisher Scientific) and then luminescence was measured using a microplate reader (GloMax^®^, Promega). Efficacy was also evaluated in OPM-2-luc+ cells. OPM-2-luc+ cells were seeded at 1.5 × 10^3^/well in a 96-well plate, and then treated with ADCs generated by mixing anti-CD63 mAb combined with linear Arg linker and Silencer Select negative control siRNA #2 (Thermo Fisher Scientific) or *luc+* siRNA at a ratio of 1:1 (1500:1500 (nM)). Luminescence activity was measured using a microplate reader (GloMax^®^, Promega), according to the manufacturer’s instructions.

### 2.4. Laser Scanning Microscopy

OPM-2-luc+ cells and NCI-H929 cells were seeded in glass-bottomed dishes (Matsunami Glass Ind, Osaka, Japan) coated with fibronectin solution (FUJIFILM Wako Pure Chemical Corporation) at 1.5 × 10^3^/well. Then, cells were treated with ADC anti-CD63 Abs with linker:F-*luc2* siRNA at the indicated ratios. Twenty-four hours after ADC treatment, cells in glass-bottomed dishes were fixed with 4% paraformaldehyde (FUJIFILM Wako Pure Chemical Corporation), then treated with rhodamine-phalloidin (100 nM, Thermo Fisher Scientific) and Hoechst 33342 (5 μM, Molecular Probe, Eugene, OR, USA) for 20 min and 5 min, respectively, at room temperature in the dark. Fluorescence images were acquired on an LSM800 laser confocal microscope (LSM) (Carl Zeiss, Jena, Germany).

### 2.5. Inhibition of ADC Uptake into MM Cells

The effects of inhibition of micropinocytosis using endocytosis inhibitors on anti-CD63 mAb-conjugated siRNA complex uptake into MM cells were investigated. OPM-2-luc cells were seeded in a 96-well plate at 1.5 × 10^4^/well. After 24 h incubation, cells were transferred to glass-bottomed dishes and then treated with endocytosis inhibitors, Rottlerin [27] (Abcam, Cambridge, UK) and Latrunculin A [28,29] (FUJIFILM Wako Pure Chemical Corporation), at 2 μM for 45 min. Then, cells were treated with anti-CD63 mAb-conjugated *luc2* siRNA labeled with FITC. Twenty-four hours after ADC treatment, OPM-2-luc cells were stained with Hoechst 33342 (5 μM) for 10 min at room temperature in the dark. Fluorescence images were acquired on an LSM800 laser confocal microscope (Carl Zeiss).

### 2.6. Quantitative Reverse Transcription-PCR (qRT-PCR)

To investigate the efficacies of Ab-conjugated siRNA complexes in suppressing targeted mRNA transcripts in MM cells, two targets, β-catenin [24,26] and c-MYC [25], were selected; we previously demonstrated that both are effective targets for MM. To target *CTNNB1* (*β-catenin*) mRNA, OPM-2-luc+ cells were cultured in a 96-well plate at 1.5 × 10^4^/well. After 24 h incubation, ADC comprising anti-CD63 mAb-branched Arg (1500 nM) and siRNA (1500 nM) was added. After 48 h of ADC treatment, RNA was purified using the NucleoSpin RNA Kit (Takara Bio Inc., Kusatsu, Japan) and subjected to reverse transcription using a ReverTra Ace^®^ qPCR RT Kit (TOYOBO Co., Ltd., Osaka, Japan). For targeting c-MYC, OPM-2-luc+ cells were seeded in a 96-well plate at 7.5 × 10^3^/well. After 24 h incubation, ADC comprising anti-CD63 mAb-branched Arg (1500 nM) and siRNA (1500 nM) was added. After 96 h treatment, RNA purification and reverse transcription were performed as described above. Human *CTNNB1*, *MYC*, and *18S ribosomal RNA* (*18S rRNA*) mRNA expression levels were measured by real-time PCR. Each real-time PCR reaction mixture contained 20 μL of TaqMan master mix (Roche Diagnostics GmbH Mannheim, Germany), cDNA, a primer pair, and a TaqMan probe (Universal Probe Library, Roche Di-agnostics GmbH). cDNA samples were amplified using a Thermal Cycler Dice system (Takara Bio, Kusatsu, Japan) with the following parameters: 95 °C for 10 min, followed by 40 cycles of 95 °C for 15 s and 60 °C for 60 s. For normalization of loading differences, *18S rRNA* mRNA was used as an internal control. The primers and TaqMan probes used in this study are shown in Appendix A.

### 2.7. Statistical Analysis

For luciferase assays, comparisons between two groups were conducted using the unpaired *t*-test. A *p*-value < 0.05 was considered statistically significant.

## 3. Results

### 3.1. Development of Ab-Conjugated siRNA Complexes

We first developed two types of Arg linkers: linear- (9mer, linear Arg) and branched- (9+9mer, branched Arg) chain polyarginine; gel shift assays were conducted using native polyacrylamide gel electrophoresis to determine the optimal ratios of anti-CD63 mAb-to-siRNA. As shown in Figure 3, the anti-CD63 mAb-linear Arg construct bound to siRNAs in a construct concentration-dependent manner, and the optimal ratio was 5:1. By contrast, the anti-CD63 mAb-branched Arg construct bound to siRNAs at lower concentrations than the former: 1:1–2.5:1. No free siRNA was observed, even when the ratio of siRNA to anti-CD63 mAb was 1:1. These observations suggest that ADCs with branched-chain Arg linkers can carry more siRNAs than those with linear-chain Arg.

### 3.2. Successful Transfer of Ab-Conjugated siRNA Complexes into Myeloma Cells

We investigated the incorporation of Ab-conjugated siRNA complexes into MM cells by examination using an LSM. First, we determined whether ADCs could transfer into adherent HeLa cells. Anti-CD63 mAb-linear Arg conjugated siRNAs was treated at a mAb-to-siRNA ratio of 5:1 (1500:300 (nM)). Confocal images showed that anti-CD63 Ab-linear Arg (9mer) conjugated with siRNAs was successfully transferred into HeLa cells (Figure 4a), and 3D images from taking Z-stack also clearly showed that these ADCs were incorporated into HeLa cells (Figure 4b,c).

We hypothesized that the branched Arg linker could bind to and deliver more siRNAs than the linear Arg linker because the positive charge for one Ab molecule with the branched Arg linker was approximately twice that with linear Arg. To test this hypothesis, we next evaluated the efficacy of branched Arg linker ADC transfer into OPM-2-luc+ MM cells, using an anti-CD63 mAb-Arg linker (both linear and branched types): F-*luc2* siRNA ratio of 5:1. Compared with a transfer of FITC-labeled siRNAs bound to ADCs with linear Arg linker into MM cells (Figure 5a), those with branched Arg linker were more effectively incorporated (Figure 5b). Furthermore, FITC-labeled siRNAs could also be transferred into NCI-H929 cells by ADC with branched Arg linkers (Appendix A). Although many siRNAs were attached to cell surfaces, Z-stack images demonstrated that siRNAs were clearly incorporated into the MM cells (Figure 5c).

Exosomes are internalized into cells via several pathways, including clathrin-mediated endocytosis [30], lipid raft-mediated endocytosis [28], receptor-mediated endocytosis [31,32], phagocytosis [29], caveolin-mediated endocytosis [33], macropinocytosis [34], and membrane fusion [35]. Therefore, we examined inhibition of exosome uptake by the endocytosis inhibitors, Rottlerin and Latrunculin. Although siRNAs conjugated with anti-CD63 mAb-branched Arg linker entered OPM-2-luc+ MM cells (Figure 6a), Rottlerin and Latrunculin inhibited Ab-siRNA conjugate incorporation. These observations suggest that siRNAs were delivered by binding to exosomes with CD63 molecules and transfer into MM cells and that this delivery was partially suppressed by endocytosis inhibition.

### 3.3. Suppression of Targeted mRNA Transcripts by Exosome-Capturing Anti-CD63 mAb-Conjugated siRNAs

Finally, we investigated the effects of anti-CD63 mAb-conjugated siRNA with branched Arg linker on targeted mRNA transcripts in MM cells by assessing the effect of anti-CD63 mAb-conjugated *luc+* siRNA complexes delivery (mAb-to-siRNA ratio, 1:1; 1500 nM) on luminescence activity. Cells were treated with Ab-conjugated siRNAs for 24 h and luciferase assays were conducted. The luminescence intensity of anti-CD63 mAb-conjugated *luc+* siRNA decreased to 55.9 ± 1.8% that of anti-CD63 mAb-conjugated control siRNA (*p* < 0.005) (Figure 7a). Next, we used qRT-PCR to examine the mRNA transcript levels of *MYC* and *CTNNB1* mRNA in OPM-2 cells treated with the anti-CD63 mAb-branched Arg construct conjugated with *MYC* and *CTNNB1* siRNAs. To determine the optimal duration of Ab-conjugated siRNA complex treatment, we exposed HeLa cells to siRNAs targeting *MYC* and *CTNNB1* using RNAiMAX (Thermo Fisher Scientific). Two siRNAs targeting *MYC* mRNA, s9129 and s9130, suppressed its transcription, while siRNA s9129 was more effective than the latter; the optimal duration for siRNA s9129 treatment was 96 h. Both siRNAs targeting *CTNNB1* (#1 and #2) also suppressed mRNA transcription, with siRNA #2 being more effective; the optimal duration for siRNA #2 treatment was 48 h (Appendix A). Treatment with anti-CD63 mAb-conjugated siRNAs at a mAb-to-siRNA ratio of 1:1 (1500 nM) decreased *MYC* and *CTNNB1* mRNA transcript levels to 52.5% and 55.3%, respectively (Figure 7b).

## 4. Discussion

Recently, exosomes have been used as a method for drug cargo delivery [14,15,16]; however, several issues need to be overcome to optimize this strategy. First, purification of exosomes: ultracentrifugation [36], polymer-based precipitation [37], size-exclusion chromatography [38], and immunoaffinity capture [39], among other methods, have been used for this purpose. Although these methods are helpful, obtaining high purity and large numbers of exosomes is time-consuming and costly. The second issue is a transfer of drugs into exosomes. Numerous transfer methods, such as electroporation [40,41] and sonication [42,43], are available; however, electroporation induces aggregation of siRNAs and the efficiency of exosome loading by electroporation is not high [44]. Furthermore, sonication is reported to induce exosome deformity [45]. To overcome these problems, direct use of exosomes as a DDS is both superior and more convenient. In accordance with our theory that the exosomes released from cancer cells are preferentially delivered into their parental cancer cells [17], we previously developed an ExomiR-Tracker, and this complex was carried into solid cancer cells and suppressed the functions of exosomal miRNAs [20]. To challenge the delivery of siRNAs into hematologic cancer cells in this study, we developed an exosome-capturing Ab-conjugated siRNA complex using a branched Arg linker (9+9mer) which is different from a linear linker (9mer) used in the ExomiR-Tracker. The results in the current experiments are limited and are partially based on previously unpublished data. However, we demonstrated that this siRNA-conjugated complex was successfully incorporated into MM cells and that this phenomenon was suppressed by endocytosis inhibitors. We also previously showed that inhibition of exosome release from cancer cells using GW4869 decreased transfer of CD63 mAb conjugated with anti-miR oligonucleotide complexes into Cal27 human oral squamous carcinoma cells and that the oligonucleotide complexes incorporated into cells by adding exosomes under a serum-free culture condition [20]. These observations demonstrate that our anti-CD63 mAb-conjugated siRNA complexes are delivered into cancer cells by uptake of the exosomes released from the parental cancer cells. Moreover, the siRNAs carried using this system successfully decreased levels of targeted mRNA transcripts.

## 5. Conclusions

Taken together, our data show that the developed exosome-capturing Ab-conjugated siRNA complexes are useful for treatment of MM cells. Importantly, as targets of siRNAs can be altered according to the optimal molecules in specific cancers, we believe that this technology will be valuable for individualized medicine. Our future work will involve investigating the pharmacokinetics and pharmacodynamics of Ab-conjugated siRNA complexes as well as assessing of their therapeutic effects in vivo. This technology has the potential to lead to a breakthrough in drug delivery systems for hematologic malignancy.

## Figures and Tables

**Figure 1 cancers-14-00566-f001:**
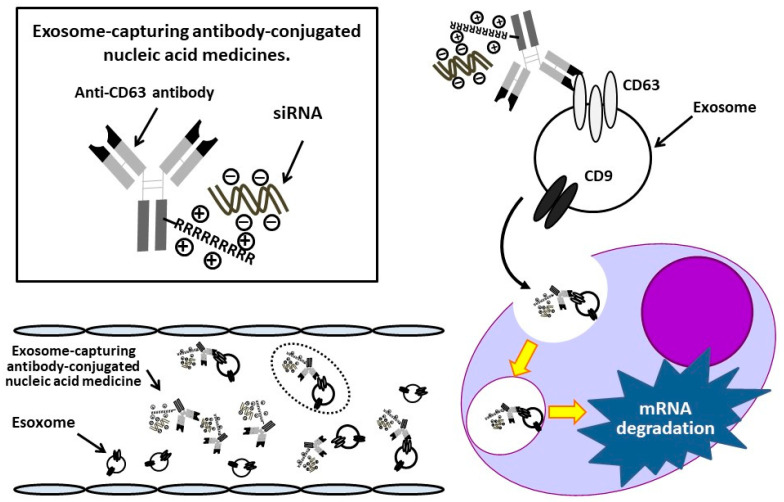
Schematic illustrating the concept of an exosome-capturing antibody-conjugated nucleic acid medicine. Antibody-conjugated siRNA complexes injected into the blood or bone marrow bind to exosomes by recognizing exosome surface antigen, CD63, and are taken up into multiple myeloma cells. Subsequently, the siRNAs bind to target mRNAs, leading to their degradation or inhibition of their translation, resulting in the suppression of myeloma cell proliferation.

**Figure 2 cancers-14-00566-f002:**
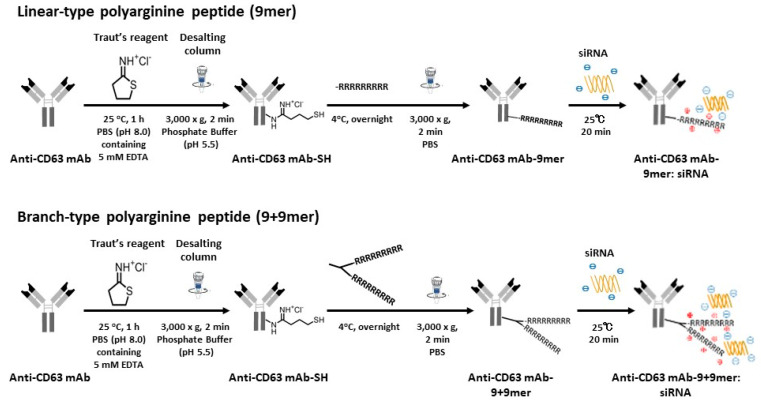
Schematic illustrating anti-CD63 mAb-conjugated siRNA complex synthesis. The procedure was conducted as described in the “Materials and methods” section. Schematic diagrams show linker-containing complex synthesis with linear arginine (9mer) (**above**) and branched arginine (9+9mer) (**below**).

**Figure 3 cancers-14-00566-f003:**
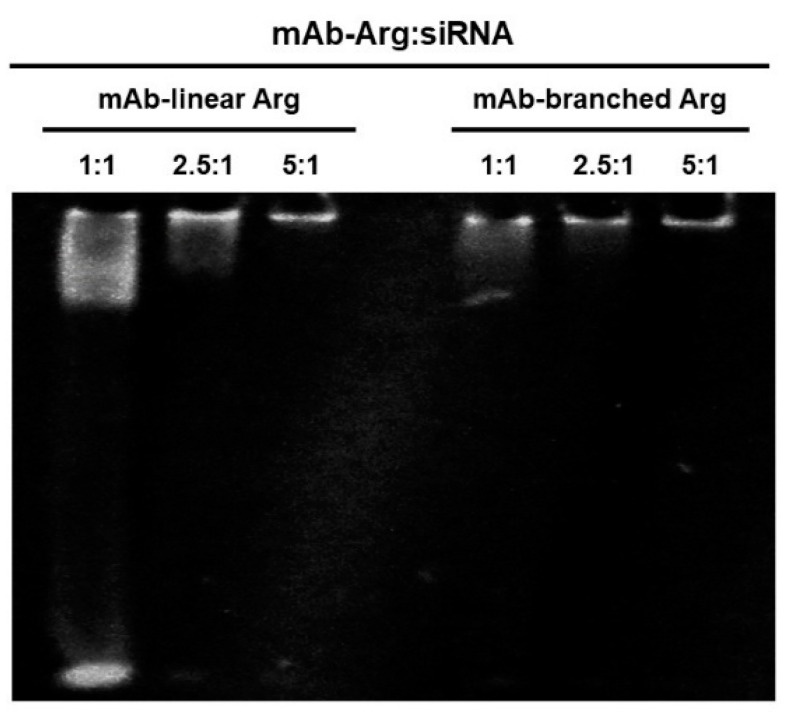
Gel shift assay. Anti-CD63 mAb-linear or -branched Arg:siRNA complexes were obtained by mixing anti-CD63 mAb bound to linear- or branched Arg linker and siRNA at various molar ratios. The lower band of the lane of mAb-linear Arg at the ratio of 1:1 indicates unbound siRNAs. (Appendix A).

**Figure 4 cancers-14-00566-f004:**
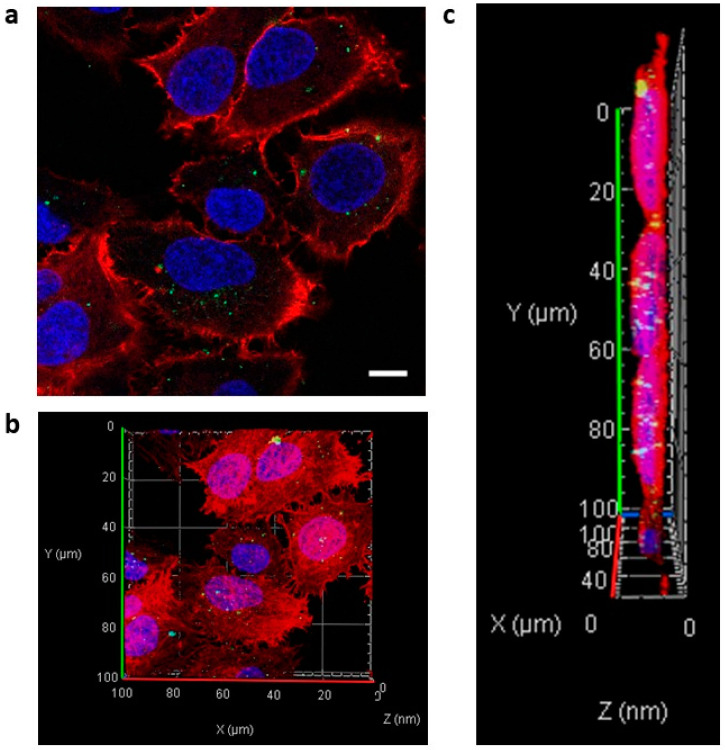
Effective transfer of anti-CD63 mAb-linear Arg conjugated with siRNA complexes into HeLa cells. The anti-CD63 mAb-Arg linear linker:siRNA ratio was 5:1 (1500:300 (nM)). Fluorescence-labeled siRNAs were successfully transferred into HeLa cells. (**a**) Confocal image of HeLa cells containing anti-CD63 mAb-linear Arg conjugated with siRNA complexes: green dots, FITC-labeled siRNAs; red, rhodamine-phalloidin; blue, Hoechst 33342. Scale bar, 20 μm. (**b**,**c**) Reconstructed Z-stack images of HeLa cells. White dots, incorporated siRNAs.

**Figure 5 cancers-14-00566-f005:**
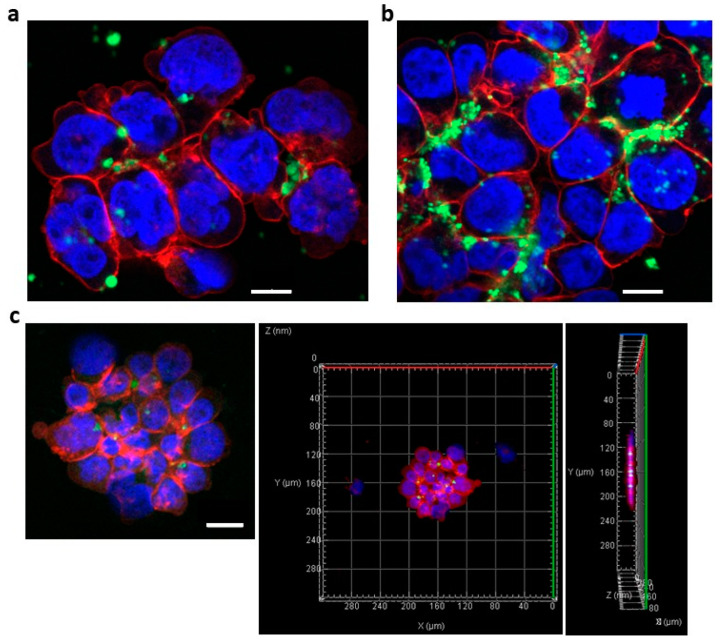
Confocal microscopic images of anti-CD63 mAb-conjugated siRNA complexes. (**a**) OPM-2-luc+ cells were treated with fluorescence-labeled anti-CD63 Ab-conjugated siRNA bound to linear Arg linkers. The anti-CD63 mAb-Arg linker:siRNA ratio was 5:1 (1500:300 (nM)). Scale bar, 10 μm. (**b**) OPM-2-luc+ cells were treated with anti-CD63 mAb-conjugated fluorescence-labeled siRNA bound to branched Arg linkers. The anti-CD63 mAb-Arg linker siRNA ratio was 5:1 (1500:300 (nM)). Scale bar, 10 μm. (**c**) Z-stack images of OPM-2-luc+ cells treated with anti-CD63 mAb-conjugated siRNA complexes. The anti-CD63 mAb-Arg linker siRNA ratio was 2.5:1 (750:300 (nM)). White dots in the right panel, incorporated siRNAs. Scale bar, 20 μm. Red, rhodamine-phalloidin; blue, Hoechst 33342 (nuclei); green, FITC-labeled siRNAs.

**Figure 6 cancers-14-00566-f006:**
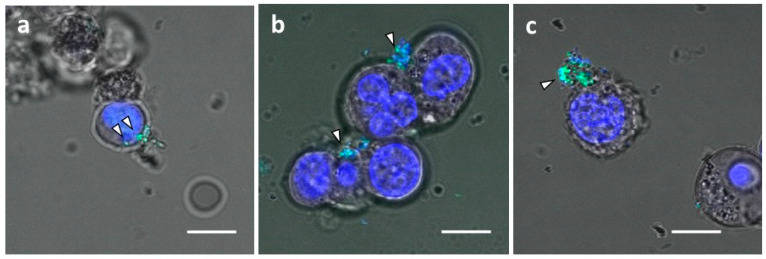
Inhibition of exosome uptake by endocytosis inhibitors. OPM-2-luc+ cells were treated with anti-CD63 Ab-conjugated FITC-labeled siRNA bound to branched Arg linkers. The anti-CD63 Ab-Arg linker:siRNA ratio was 2.5:1 (750:300(nM)). (**a**) Vehicle treatment. (**b**) Latrunculin treatment (2 μM). (**c**) Rottlerin treatment (2 μM). Arrowheads indicate anti-CD63 Ab-conjugated FITC-labeled siRNA complexes. Scale bar, 10 μM. Blue, Hoechst 33342 (nuclei); green, FITC-labeled siRNAs.

**Figure 7 cancers-14-00566-f007:**
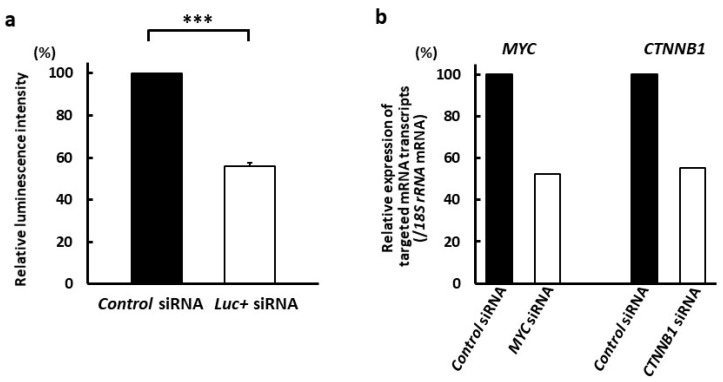
Successful suppression of targeted molecules in MM cells by anti-CD63 mAb-conjugated siRNA complexes. (**a**) Luminescence activity was significantly decreased in OPM-2 MM cells treated with *luc+* siRNAs compared with in cells treated with control siRNAs (*** *p* < 0.005). The anti-CD63 mAb-Arg linker:siRNA ratio was 1:1 (1500 nM). Data are presented as mean ± standard error of two independent experiments (four wells per experiment). (**b**) The targeted mRNA transcripts decreased in OPM-2 MM cells treated with *MYC* or *CTNNB1* siRNAs compared with those in cells treated with control siRNA. The anti-CD63 mAb-Arg linker:siRNA ratio was 1:1 (1500 nM). Data show the representative of two independent experiments.

## Data Availability

The data used in the current study are available from the corresponding author upon reasonable request. The data are not publicly available due to patent application.

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
