# Peer review of "Successful Incorporation of Exosome-Capturing Antibody-siRNA Complexes into Multiple Myeloma Cells and Suppression of Targeted mRNA Transcripts"

_cancers, 2022, doi:10.3390/cancers14030566_

Round 1

Reviewer 1 Report

The authors reported on the CD69/oligo-arginine conjugate as a siRNA delivery system that are based on the exosome-mediated tumor homing. While the strategy of exosome-capturing antibody seems unique, the data set and experimental design does not support it. Also, the strategy is likely to be used only for the xenograft mouse model.

Correctively, the manuscript is not suit to be accepted in present form.

  1. The most critical point is that the authors showed only the in vitro data. Thus, even though the cellular uptake and gene knockdown effect was shown, it does not support the exosome-mediated cancer targeting since anti-CD63 antibody can directly bind to the cellular surface.
  2. The authors used the human CD63-targeting antibody for the targeting of human cells. However, it is plausible that the CD63-expressing exosomes exist in the plasma because it is a typical marker for the exosome. Thus, the CD63 antibody can bind to the exoxome non-specifically, in parallel with tumor-oriented/homing exosomes. While the human antibody can specifically recognize the tumor-oriented exosome in xenograft mouse model, this strategy might be not applicable in patient.
  3. Overall, the “multivesicular body” should be used instead of “exosome”.
  4. In Fig 7b, triplicate experiments with statistical analysis is needed.

Author Response

Response to Reviewer #1

The authors reported on the CD69/oligo-arginine conjugate as a siRNA delivery system that are based on the exosome-mediated tumor homing. While the strategy of exosome-capturing antibody seems unique, the data set and experimental design does not support it. Also, the strategy is likely to be used only for the xenograft mouse model.

              We wish to express our appreciation to the reviewer for the insightful comments, which have helped us to improve the manuscript and our next projects significantly.

Comments:

  1. The most critical point is that the authors showed only the in vitro data. Thus, even though the cellular uptake and gene knockdown effect was shown, it does not support the exosome-mediated cancer targeting since anti-CD63 antibody can directly bind to the cellular surface.

       Thank you very much for your comment. I apologize that the reviewer might have been confused. Anti-CD63 monoclonal antibody (mAb) conjugated with siRNAs binds to exosomes (not to cellular surface). The exosomes which are bound with the anti-CD63 mAn-conjugated siRNAs bind and incorporate into cancer cells. This process was partially suppressed by endocytosis inhibition as shown in Figure 6. Moreover, in our previous publication in Pharmaceutics (12:545, 2020) (Reference #20), we demonstrated that anti-CD63 mAb-conjugated oligonucleotide complexes were incorporated into cells exosome-dependently. The uptake of these complexes was inhibited by GW4869, an inhibitor of exosome generation (We have described this finding in the section of Discussion of the first version, Lines305-308). We also demonstrated that exosome-dependent cellular uptake of anti-CD63 mAb under a serum-free culture condition. We added this observation in the section of discussion in the revised manuscript (Lines 308-309). Taken together, these findings support that anti-CD63 mAb-conjugated siRNA complexes show exosome-mediated cancer targeting.

  1. The authors used the human CD63-targeting antibody for the targeting of human cells. However, it is plausible that the CD63-expressing exosomes exist in the plasma because it is a typical marker for the exosome. Thus, the CD63 antibody can bind to the exosome non-specifically, in parallel with tumor-oriented/homing exosomes. While the human antibody can specifically recognize the tumor-oriented exosome in xenograft mouse model, this strategy might be not applicable in patient.

       Thank you very much for your important comments. I appreciate deeply. We have previously demonstrated that tumor cell-derived exosomes show a tropism and are preferentially internalized by tumor cells in an in vitro experiment ( Reference #17). Based on this finding, we conducted the current experiments and demonstrated the suppression of oncogenes in MM cells. Therefore, we believe that the anti-CD63 mAb-conjugated siRNAs capture MM cell-derived exosomes and incorporate them into MM cells in a human setting. However, this should be clarified in clinical studies and we will conduct Phase 0 using radioisotope-labeled siRNAs to answer the reviewer’s comment in the future. I appreciate the important comment again.

  1. Overall, the “multivesicular body” should be used instead of “exosome”.

       Thank you very much for your comment. But I don’t agree with the reviewer’s comment. Multivesicular bodies produce exosomes in their inner site during their maturation and dock at the cell surface. Then, exosomes are released extracellularly via exocytosis of multivesicular bodies. Therefore, “exosome” is correct.

  1. In Fig 7b, triplicate experiments with statistical analysis are needed.

       We appreciate your comment deeply. However, we, unfortunately, cannot resume these experiments because Ms. Soma, the first author of this manuscript, is my undergraduate student and has to study to prepare for the National Examination for Pharmacist this February. And we have moved to the next project to investigate the pharmacokinetics/pharmacodynamics of this anti-CD63 mAb-conjugated siRNA complex with other collaborators. We have confirmed the similarly inhibitory effects by two independent experiments.

Reviewer 2 Report

The authors do an excellent job presenting and articulating compelling evidence for the mechanism of action for exosome-capturing Ab-conjugated siRNA complexes.  The scientific rationale is sound, and the evidence presented appears to support their claims. I would accept the paper in its current format. 

Author Response

The authors do an excellent job presenting and articulating compelling evidence for the mechanism of action for exosome-capturing Ab-conjugated siRNA complexes.  The scientific rationale is sound, and the evidence presented appears to support their claims. I would accept the paper in its current format.

              We deeply appreciate that the reviewer highly estimated our works and accepted this manuscript.

Reviewer 3 Report

In their manuscript the authors show a new potential method of delivery of anticancer agents. An exosome-capturing Ab-conjugated siRNAs with branched Arg linkers were shown to be potentilally effectively delivered into MM cells via uptake of exosomes by parental cells. In overall the manuscript is well written and the obtained data adeqately presented. The work is novel and interesting, however some additional data should be presented:

1) what is the efficacy of the proposed compund production? How is the effiacy of the process measured? I do not see such data in the manuscript.

2) does the conjugated siRNA is characetezied by increased stability?

3) Please provide controls for the data provided in Figure 3

4) In what concentration/amounts was the obtained compound used in the experiments described in sections 3.2 and 3.3

5) Does sole anti-CD63 or not conjugated siRNA have any effect on the results of experiments described in secion 3.2 and 3.3. Please provide a commentary and respective controls.

6) 

Author Response

In their manuscript, the authors show a new potential method of delivery of anticancer agents. An exosome-capturing Ab-conjugated siRNAs with branched Arg linkers were shown to be potentially effectively delivered into MM cells via uptake of exosomes by parental cells. In overall, the manuscript is well written and the obtained data is adequately presented. The work is novel and interesting, however some additional data should be presented

              We wish to express our appreciation to the reviewer for the insightful comments, which have helped us to improve the manuscript and our next projects significantly. We appreciate your comments deeply.

              However, we unfortunately cannot resume these experiments because Ms. Soma, the first author of this manuscript, is my undergraduate student and has to study to prepare for the National Examination for Pharmacist this February. And we have moved to the next project to investigate the pharmacokinetics/pharmacodynamics of this anti-CD63 mAb-conjugated siRNA complex with other collaborators. Therefore, we respond to the reviewer’s comments as described below:

  • what is the efficacy of the proposed compound production? How is the efficacy of the process measured? I do not see such data in the manuscript.

        Thank you very much for the important comment. We deeply appreciate it. However, we did not unfortunately measure the efficacy of the compound production nor the efficacy of the process in the current study. I apologize that we cannot answer this comment by the reviewer this time. However, I planned to measure the efficacy in the ongoing experiments on pharmacodynamics/pharmacokinetics of this proposed complex. I will be able to answer the reviewer’s comment in the future. I deeply appreciate the comment again.

  • Is the conjugated siRNA characterized by increased stability?

        Thank you very much for the important comment. We deeply appreciate it. Unfortunately, we did not evaluate the stability of the conjugated siRNAs in the serum. However, we succeeded in the incorporation of unmodified and fluorescence-labeled siRNAs into cells and the inhibition of the targeted-mRNA transcript levels in MM cells by LNA and2’-fluoro-modified siRNAs. These findings show that these siRNA complexes are stable enough to work in the serum. However, to improve the efficacy of inhibition by the siRNA complex, we have to estimate the stability in the serum. We will estimate it in the future study. I appreciate this important comment deeply.

  • Please provide controls for the data provided in Figure 3

        Thank you very much for your comment. In Figure 3, we demonstrate the differences of bands (or smears) in accordance with the various molar ratios of anti-CD63 mAb-to-siRNA. I apologize that we did not use the control in the current study. We have previously demonstrated Gel Shift assay in Figure S4 in Pharmaceutics (12:545, 2020) (Reference #20). As in the previous experiment, the lower band of the lane of mAb-linear Arg at the ratio of 1:1 indicates unbound siRNAs. In the legend of Figure 3, we added the sentence that the lower band of the lane of mAb-linear Arg at the ratio of 1:1 indicates unbound siRNAs (Lines 207-208). I appreciate your comment.

  • In what concentration/amounts was the obtained compound used in the experiments described in sections 3.2 and 3.3?

        Thank you very much for your comment. In Figure 4, we treated Anti-CD63 mAb-linear Arg conjugated siRNAs at a mAb-to-siRNA ratio of 5:1 (1500:300 [nM]), but I did not describe it. I did not state the concentration of the complex in the legend of Figure 7 although I described it in the section of Results regarding Figure 7. I deeply apologize to them. I added the concentration of the complexes used in the experiments in the section of Results (Lines 212-213) and the legends of Figures 4 (Lines 218-219) and 7 (Lines 218 and 284) using the Word editing device.

  • Does sole anti-CD63 or not conjugated siRNA have any effect on the results of experiments described in sections 3.2 and 3.3? Please provide a commentary and respective controls.

        Thank you very much for your comment. I deeply apologize that we cannot show the data the reviewer requested. But in our previously unpublished data, neither treatment of sole anti-CD63 mAb nor unconjugated siRNA affected the levels of mRNA transcripts. It is suggested that either treatment of sole anti-CD63 mAb or unconjugated siRNA would never suppress the levels of mRNA transcripts or incorporate them into cells.

  • About the comment of “English language and style”

        In the comment, the reviewer mentioned that the manuscript is well written. I appreciate this comment. However, the reviewer marked “English language and style are fine/minor spell check required”. In accordance with the reviewer’s comment, I corrected several sentences (Lines 262, 275-276, and 279).

        I’ve requested Bioedit (https://www.bioedit.com/services) to edit our manuscript before submission and an English native scientist who is a specialist in this field edited it. Then, we have rewritten it in accordance with his edition. We also checked the spell and style very carefully. If the reviewer considers that it is necessary to rewrite more, I would like to ask the reviewer to point out where the expression is to be corrected. We appreciate the cooperation of the reviewer.

Round 2

Reviewer 1 Report

Now acceptable

Author Response

Response to Reviewer #1

Now acceptable

We deeply appreciate that the reviewer decided to accept this manuscript.

Reviewer 3 Report

As most of the issues were not solved in the revised version of the manuscript please incorporate an addiational paragraph in the disccussion section refrering to the current limitations of the data and previous experiments using the presented construct.

Author Response

Response to Reviewer #3

As most of the issues were not solved in the revised version of the manuscript, please incorporate an additional paragraph in the discussion section referring to the current limitations of the data and previous experiments using the presented construct.

Thank you very much for your kind and appropriate comment. I deeply appreciate it. I apologize that our work is limited and is partially based on the previous unpublished data in the present construct. As I have no idea where we should incorporate an additional paragraph in the Discission section, I added a comment in the Discussion section as your suggestion (Lines 304-305) written in red with yellow highlighted. I wish you would kindly accept this manuscript. Thank you very much again for your review.